

# Merged SAGE II, Ozone_cci and OMPS ozone profiles dataset and evaluation of ozone trends in the stratosphere

Viktoria F. Sofieva[1], Erkki Kyrölä[1], Marko Laine[1], Johanna Tamminen[1], Doug Degenstein[2], Adam Bourassa[2], Chris Roth[2], Daniel Zawada[2], Mark Weber[3], Alexei Rozanov[3], Nabiz Rahpoe[3], Gabriele Stiller[4], Alexandra Laeng[4], Thomas von Clarmann[4], Kaley A. Walker[5], Patrick Sheese[5], Daan Hubert[6], Michel van Roozendael[6], Claus Zehner[7], Robert Damadeo[8], Joseph Zawodny[8], Natalya Kramarova[9,10], Pawan K. Bhartia[9]

[1] Finnish Meteorological Institute, Helsinki, Finland
[2] Institute of Space and Atmospheric Studies, University of Saskatchewan, Saskatoon, Canada
[3] Institute for Environmental Physics, University of Bremen, Bremen, Germany
[4] Karlsruhe Institute of Technology, Institute of Meteorology and Climate Research, Karlsruhe, Germany
[5] Department of Physics, University of Toronto, Toronto, Canada
[6] Royal Belgian Institute for Space Aeronomy (BIRA-IASB), Brussels, Belgium
[7] ESA/ESRIN, Frascati, Italy
[8] NASA Langley Research Center, Hampton, VA, USA
[9] NASA Goddard Space Flight Center, Silver Spring, MD, USA
[10] Science Systems and Applications Inc., Lanham, MD, USA

*Correspondence to*: Viktoria Sofieva (viktoria.sofieva@fmi.fi)

**Abstract.** In this paper, we present a merged dataset of ozone profiles from several satellite instruments: SAGE II on ERBS, GOMOS, SCIAMACHY and MIPAS on Envisat, OSIRIS on Odin, ACE-FTS on SCISAT, and OMPS on Suomi-NPP. The merged dataset is created in the framework of European Space Agency Climate Change Initiative (Ozone_cci) with the aim of analyzing stratospheric ozone trends. For the merged dataset, we used the latest versions of the original ozone datasets. The datasets from the individual instruments have been extensively validated and inter-compared; only those datasets, which are in good agreement and do not exhibit significant drifts with respect to collocated ground-based observations and with respect to each other, are used for merging. The long-term SAGE-CCI-OMPS dataset is created by computation and merging of deseasonalized anomalies from individual instruments.

The merged SAGE-CCI-OMPS dataset consists of deseasonalized anomalies of ozone in 10° latitude bands from 90°S to 90°N and from 10 to 50 km in steps of 1 km covering the period from October 1984 to July 2016. This newly created dataset is used for evaluating ozone trends in the stratosphere through multiple linear regression. Negative ozone trends in the upper stratosphere are observed before 1997 and positive trends are found after 1997. The upper stratospheric trends are statistically significant at mid-latitudes in the upper stratosphere and indicate ozone recovery, as expected from the decrease of stratospheric halogens that started in the middle of the 1990s.





# 1    Introduction

The ozone layer protects life on Earth from harmful ultraviolet solar radiation and plays an important role in the radiation budget of the atmosphere. Its evolution is intimately coupled to climate change. Starting in the 1970s, the stratospheric ozone declined worldwide, with the largest decline of 4 to 8 % per decade seen in the upper stratosphere

(WMO, 2014; Steinbrecht et al., 2017, and references therein). After international agreements on the reduction of ozone-depleting substances (Montreal Protocol and its Amendments), the first signatures of ozone recovery were observed, at least in certain latitude and altitude regions (e.g., Newchurch et al., 2003; Kyrölä et al., 2013; Bourassa et al., 2014; Eckert et al., 2014; Harris et al., 2015; Tummon et al., 2015; WMO, 2014). About half of the recent increase in upper stratospheric ozone is attributed to climate change (WMO, 2014).  As ozone recovery will remain strongly influenced by climate change,

continued monitoring of stratospheric ozone and its vertical structure is important.

The European Space Agency (ESA) Ozone Climate Change Initiative (Ozone_cci) aims to generate new high-quality satellite data sets that are essential to assess the fate of atmospheric ozone and better understand its link with anthropogenic activities (http://www.esa-ozone-cci.org). Participating in the Ozone_cci project are three instruments on board Envisat, GOMOS (Global Ozone Monitoring by Occultation of Stars), MIPAS (Michelson Interferometer for Passive Atmospheric

Sounding) and SCIAMACHY (SCanning Imaging Spectrometer for Atmospheric CHartographY), as well as OSIRIS (Optical Spectrograph and InfraRed Imaging System) and SMR (Sub-Millimeter Radiometer) both aboard Odin and ACE-FTS (Atmospheric Chemistry Experiment Fourier Transform Spectrometer) on SCISAT.

Satellite data provide good spatial coverage, but the temporal coverage of their data records is usually too short for trend analyses. For reliable estimates of ozone trends long-term data records are needed in order to separate natural ozone

variability (e.g., due to solar activity) and long-term trends. For the assessment of ozone trends, several merged ozone datasets with a good vertical resolution have been created (Harris et al., 2015; Tummon et al., 2015): GOZCARDS (Froidevaux et al., 2015), SWOOSH (Davis et al., 2016), SAGE II – OSIRIS (Bourassa et al., 2014) and SAGE II – GOMOS (Kyrölä et al., 2013). In construction of these datasets, different merging approaches are used. GOZCARDS and SWOOSH provide ozone mixing ratios on pressure levels, while SAGE II – OSIRIS and SAGE II – GOMOS datasets provide number

density on a geometric altitude grid as used in the respective data retrievals.

This paper introduces a new dataset in which reliable satellite data providing ozone profiles on an altitude grid are merged into a climate data record for assessment of ozone trends. These are from the five Ozone_cci instruments: GOMOS, MIPAS, SCIAMACHY, OSIRIS and ACE-FTS covering the period 2001- 2016, which are merged with the data from SAGE II (Stratospheric Aerosol and Gases Experiment II, 1984-2005) and OMPS-LP (Ozone Monitor Profiling Suite-Limb

Profiler, 2012-2016). Since the last ozone assessment (WMO, 2014), the Ozone_cci satellite data were processed with new retrieval versions. The stability of the individual instrument data records has been extensively studied; only sufficiently stable data are used for the merged dataset. The merging is performed on deseasonalized anomalies computed from each individual dataset. This method is often used for creating a long-term data records (e.g., IPCC, 2013; WMO, 2014), as well



as for trend analyses of ozone (e.g., Bourassa et al., 2014; Randel and Thompson, 2011; Sioris et al., 2014; Steinbrecht et al., 2017), stratospheric temperature (Randel et al., 2009; Seidel et al., 2011; Thompson et al., 2012) and water vapor (Jones et al., 2009). The main advantage of using deseasonalized anomalies is that biases due to different sampling patterns (including the difference in local time) and instrumental biases are automatically removed, if the sampling patterns do not change over

time. While assessing the trends using deseasonalized anomalies, there is no need to fit the seasonal variations with harmonic functions (the seasonal variations do not necessary allow a simple expansion into a few harmonics).

The paper is organized as follows. Section 2 describes the ozone datasets from the individual instruments. Section 3 is dedicated to comparisons of the individual datasets and evaluation of the deseasonalized anomalies. In Section 4 we describe the merging method and associated uncertainties. Section 5 is dedicated to evaluation of ozone trends in the stratosphere

using the merged SAGE-CCI-OMPS dataset. The information about data availability is provided in Section 6 and the conclusions are summarized in Section 7.

## 2    Data

Ozone measurements available from different satellite instruments cover all seasons and various times of day, as well they have good latitudinal coverage. The information about individual datasets is collected in Table 1. All of the data used

for creating the merged dataset have a sufficiently good vertical resolution of 1–3 km in the stratosphere and in the UTLS (upper troposphere and the lower stratosphere). For all instruments used here, ozone profiles are retrieved on the geometric altitude grid. The majority of the datasets − SAGE II, GOMOS, OSIRIS, SCIAMACHY and OMPS − provide number density ozone profiles; therefore this representation is adopted for the merged dataset. For ACE-FTS and MIPAS, the retrievals are in volume mixing ratio on an altitude grid. Conversion to number density profiles is performed using

temperature profiles retrieved by these instruments, thus providing consistent (i.e., without using external information about temperature and pressure profiles) representation of number density ozone profiles. Since the publication of the WMO 2014 ozone assessment, new processing versions of the Ozone_cci datasets were introduced. Generally, the newly reprocessed ozone datasets have smaller biases and are more stable (further details are provided in the descriptions of individual datasets below).

The time series of the number of available ozone profiles per month from each instrument is shown in Figure 1. Note that for some instruments, the selected time period is shorter than the full operation period (Table 1). The individual datasets have been compared with each other and with ground-based data and only time periods when the instruments were operating optimally are selected for merging (details are given later in Section 3).

The Ozone_cci datasets, which are used for merging, are included in the user-friendly HARMonized dataset of OZone

profiles (HARMOZ) (Sofieva et al., 2013) and are available at the Ozone_cci web-page (http://www.esa-ozone-cci.org/?q=node/160). HARMOZ consists of the original retrieved ozone profiles from each instrument, which are screened



for invalid data by the instrument experts and are presented on a vertical grid and in a common netCDF4 format, which simplifies the data usage. Below are more detailed descriptions of the individual datasets.

**Table 1. Information about the datasets used in the merged dataset**

| Instrument/ satellite | Processor, data source | Time period | Local time | Vertical resolution | Estimated precision | Profiles per day |
|---|---|---|---|---|---|---|
| SAGE II/ ERBS | NASA V7.0, original files | Oct 1984 – Aug 2005 | sunrise, sunset | ~1 km | 0.5-5% | 14-30 |
| OSIRIS/ Odin | USask v 5.10, HARMOZ_ALT | Nov 2011 – July 2016 | 6 a.m., 6 p.m. | 2-3 km | 2-10% | ~250 |
| GOMOS/ Envisat | ALGOM2s v 1.0, HARMOZ_ALT | Aug 2002 – Aug 2011 | 10 p.m. | 2-3 km | 0.5–5 % | ~110 |
| MIPAS/ Envisat | KIT/IAA V7R_O3_240, HARMOZ_ALT | Jan 2005 – Apr 2012 | 10 p.m., 10 a.m. | 3-5 km | 1–4% | ~1000 |
| SCIAMACHY/ Envisat | UBr v3.5, HARMOZ_ALT | Aug 2003- Apr 2012 | 10 a.m. | 3-4 km | 1-7% | ~1300 |
| ACE-FTS/ SCISAT | v3.5/3.6, HARMOZ_ALT | Feb 2004 – Dec 2016 | sunrise, sunset | ~3 km | 1-3% | 14-30 |
| OMPS/ Suomi NPP | USask 2D, HARMOZ_ALT | Apr 2012- Aug 2016 | 1:30 p.m. | ~1 km | 2-10% | ~1600 |

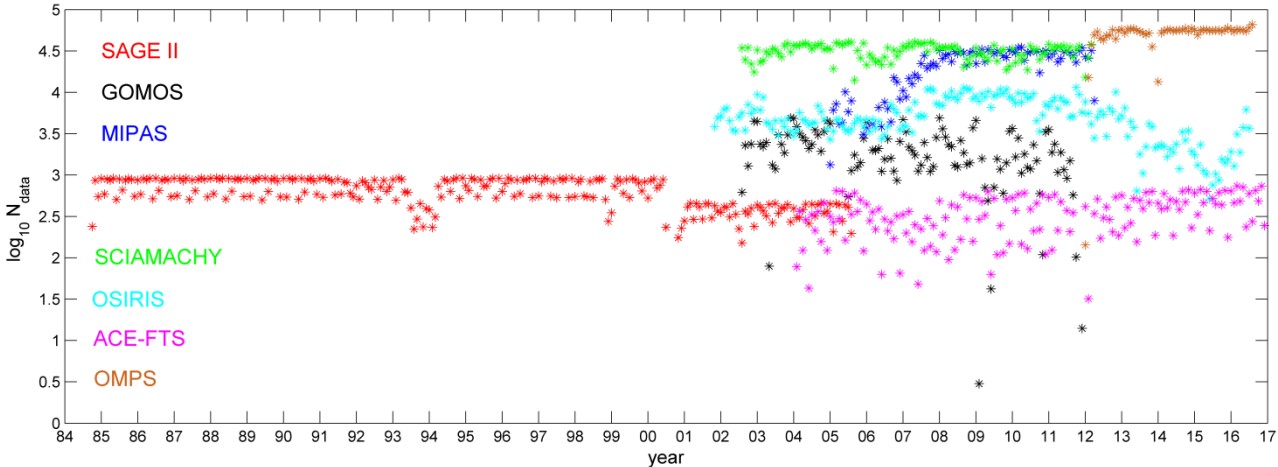

**Figure 1. Monthly data volume (logarithm of number of measurements).**

## 2.1 SAGE II

SAGE II operated on board the Earth Radiation Budget Satellite (ERBS) from 1984 to 2005. Using the solar occultation technique to observe the Sun during sunrises and sunsets, SAGE II observed the atmosphere in seven channels with wavelengths between 375 and 1030 nm (Mauldin III et al., 1985). Vertical slant-path transmission profiles with a 1 km resolution were inverted into profiles of ozone, aerosol extinction, water vapor, and nitrogen dioxide using a simple 'onion-peeling' method (Chu et al., 1989). Ozone is inferred primarily from spectral measurements near 600 nm at the peak of the



Chappuis band and the resulting high-quality profiles have random uncertainty < 1% in the stratosphere (McCormick et al., 1989). This work uses version 7.00 of the SAGE II data set (Damadeo et al., 2013) and have been filtered for invalid values as described in Damadeo et al. (2014). The data corresponding to heavy aerosol loading are excluded as recommended by Wang et al. (2002).

Due to the self-calibrating nature of occultation measurements, SAGE II data are stable (Hubert et al., 2016) and have been thoroughly validated (e.g., Damadeo et al., 2013; McCormick et al., 1989; Wang et al., 2002) and used in previous ozone assessments (e.g., WMO, 2011). Given its quality and stability SAGE II data are also included in several merged data sets that have been used in the most recent ozone assessment (WMO, 2014). Since SAGE II offers one of only a few data sets during the 1990s or earlier, it is often the only source of data prior to about 2002 incorporated into many merged data
sets.

## 2.2   GOMOS

GOMOS is a stellar occultation instrument that operated on board of Envisat in 2002-2012 (Bertaux et al., 2010; Kyrölä et al., 2010). Ozone profiles are retrieved from the ultraviolet (UV) and visible spectrometer measurements at wavelengths between 250 and 692 nm. The main dataset consists of nighttime ozone profiles (with solar zenith angle larger
than 105°), which are retrieved from atmospheric transmittance spectra. There also exists the bright-limb ozone profile dataset derived from the GOMOS limb scattering measurements (Tukiainen et al., 2011, 2015). However, the altitude range for the bright-limb ozone profiles is limited, and there are some indications of a drift of retrieved ozone concentrations. Therefore, only night-time GOMOS ozone profiles are used for the merged dataset. The GOMOS ozone profiles are obtained with the ALGOM2s v1.0 processor (Sofieva et al., 2017). ALGOM2s is nearly identical to the ESA IPF v6
processor (GOMOS IPF v6 data were used in the WMO-2014 ozone assessment) in the stratosphere, but has improved data quality in the UTLS.

GOMOS provides stratospheric ozone profiles with a vertical resolution of 2 km below 30 km, 3 km above 30 km, with a linear transition between (Tamminen et al., 2010). The vertical resolution of the GOMOS ozone profiles is the same for all occultations due to the Tikhonov-type (Tikhonov, 1963) target-resolution regularization (Kyrölä et al., 2010; Sofieva
et al., 2004). The stellar flux recorded by GOMOS, and thus signal-to-noise ratio and precision of retrieved profiles, depends on stellar magnitude and spectral class. The estimated random uncertainty of GOMOS ozone profiles in the stratosphere is 0.5-5 % (Tamminen et al., 2010). Validation of estimated uncertainties for ozone profiles in the stratosphere has shown that they are realistic except for cases when ozone profiles are derived from occultations of dim stars (Sofieva et al., 2014b).

The validation and intercomparison results have shown that the GOMOS night-time ozone profiles have small biases
with respect to ground-based measurements (Hubert et al., 2016). GOMOS profiles are in a very good agreement with SAGE II and OSIRIS measurements in the stratosphere (Adams et al., 2014; Hubert et al., 2016; Kyrölä et al., 2013), as well as with OSIRIS, MIPAS and ACE-FTS measurements in the UTLS (Sofieva et al., 2017).





The GOMOS exploits a self-calibrating measurement principle, therefore a high stability of the GOMOS data is expected (Kyrölä et al., 2010). It turned out that it is important to exclude the ozone data from the stars with insufficient UV-flux. These data are biased and can induce artificial data drifts. GOMOS IPF V6 data have been combined into the merged SAGE II-GOMOS dataset, which was used for ozone trend analysis (Harris et al., 2015; Kyrölä et al., 2013; Laine et al.,

2014; Tummon et al., 2015; WMO, 2014). The new GOMOS ALGOM2s dataset used for the merged dataset not only has improved data quality in the UTLS, but it is also expected to be more stable in the whole atmosphere due to an advanced screening of unreliable data (Sofieva et al., 2017).

## 2.3   MIPAS

The Michelson Interferometer for Passive Atmospheric Sounding (MIPAS) is an infrared limb emission spectrometer

that was flown on the Envisat platform (Fischer et al., 2008). In 2002-2004, the instrument operated at full spectral resolution. Due to a failure of the instrument's mirror slide in 2004, the operations were suspended for almost a year and were resumed in 2005 with reduced spectral, but improved vertical resolution. These operations continued until the loss of communications with the ENVISAT platform in April 2012.

Stratospheric ozone profiles aren retrieved from MIPAS/ENVISAT limb emission spectra. In this work, we use the

scientific MIPAS processor V7R_O3_240 developed at Karlsruhe Institute of Technology, IMK/IAA V7R_O3_240. The retrieval is performed via constrained inverse modelling of limb radiances. In the stratospheric/tropospheric retrievals, local thermodynamic equilibrium (LTE) is assumed. A detailed description can be found in von Clarmann et al. (2003; 2009). The data version used in this work is retrieved from new level 1 spectra (version V7), in which a new set of (time-dependent) correction coefficients for the non-linearity in the detectors' response functions is implemented. This has a major positive

impact on the stability of the dataset (Laeng et al., 2017). The main updates in the retrieval strategy include the following. The retrieval of temperature, which is crucial for subsequent trace gas retrievals, has been revised. The atmospheric background continuum radiation is no longer set to zero above 30 km but fitted also for higher altitudes. The treatment of interfering $H_2O$ lines has been improved; and two additional microwindows are used (see Laeng et al. (2017) for details).

Due to their different data characteristics, the two MIPAS measurement periods are usually treated as two independent

datasets. Their processing schemes are different, and the vertical resolution of the early MIPAS period is lower than that of the later period: 3.5–6 km vs 2–5 km for retrieved ozone. The vertical resolution of MIPAS early period is also worse than the vertical resolution of all of the other datasets used in the merged dataset. Instead of downgrading the vertical resolution of all other participating datasets to that of the MIPAS early period, which is also very short, the MIPAS early period data were discarded from the merging. The mean uncertainty of retrieved MIPAS ozone profiles in 2005-2012 is 0.05-0.1 ppmv

30   (1-5 %).



## 2.4 SCIAMACHY

SCIAMACHY aboard ENVISAT was a European space-borne spectrometer measuring the upwelling radiation from the Earth's atmosphere in the UV, visible, near and shortwave infrared spectral ranges. The instrument provided measurements in nadir, limb, and solar/lunar occultation viewing geometries. In the limb viewing geometry, the SCIAMACHY instrument scanned the Earth's atmosphere vertically from about 3 km below the horizon (0 km after October 2010) with a vertical sampling of about 3.3 km and a vertical instantaneous field of view of 2.6 km (at tangent height). At each tangent height, a horizontal scan within a total swath of 960 km was performed. A detailed description of the instrument and its measurement modes is given by Burrows et al. (1995) and Bovensmann et al. (1999).

This study uses V3.5 of the SCIAMACHY limb ozone retrieval, which is a completely new retrieval as compared to V2.9 used e.g. by Sofieva et al. (2013). The V3.5 SCIAMACHY retrieval uses six spectral windows (264 - 265 nm, 266.5 - 267.7 nm, 272.5 - 273.8 nm, 276.5 - 278 nm, 282.5 - 284 nm, 289 - 309.5 nm) in the Hartley ozone absorption band as well as one window (325 - 331 nm) in the Huggins band and one window (495 - 576 nm) in the Chappuis ozone band. The radiances in the Huggins and Chappuis absorption bands are sun-normalized while normalization to an upper tangent height is used in the Hartley absorption band. To reduce the influence of calibration errors and broad-band spectral features, a polynomial is subtracted from the logarithms of the normalized radiances in all spectral windows except for the first three. The zero order polynomial (i.e. a constant) is subtracted from the measurements in the last three windows in the Hartley band (a common polynomial is calculated for the fourth and fifth spectral windows), while a linear and a quadratic polynomials are subtracted in the Huggins and Chappuis bands, respectively. Independent values for the surface albedo in the UV and visible spectral ranges are retrieved simultaneously with the ozone number density.

Ozone is retrieved from 8 to 60 km on the measurement tangent height grid. Tangent heights with clouds and/or highly increased aerosols in the field of view are rejected and the ECSTRA (Extinction Coefficient for STRatospheric Aerosol) database (Bingen and Fussen, 2000; Fussen and Bingen, 1999) is used as the aerosol model (based on SAGE II climatology). The random error for V3.5 is on the order of 1-5%, lower than that reported in Rahpoe et al. (2013) for an earlier version. The pointing accuracy is similar to previous versions (see Sofieva et al., 2013). The vertical resolution of the SCIAMACHY profiles is about 3.3 km. Intercomparison of V3.5 SCIAMACHY limb ozone with Microwave Limb Sounder (MLS) and ozonesonde data shows agreement generally within 5% (paper in preparation).

## 2.5 OSIRIS

OSIRIS (Optical Spectrograph and InfraRed Imaging System) is a Canadian instrument on board the Swedish satellite Odin that was launched in February of 2001. It is a limb-viewing device that makes repeated measurements of the limb scattered radiance in the UV and visible spectral ranges with a sampling of approximately 2 km between 10 km and 100 km of altitude. OSIRIS uses limb radiance spectra to generate ozone profiles in a range from 80° S to 80° N. Concentrations are retrieved on a 1 km grid from 10.5 km to 59.5 km and have a vertical resolution of ~2 km. The ozone retrieval by the





University of Saskatchewan uses a multiplicative algebraic reconstruction technique as described in Degenstein et al. (2009). The OSIRIS instrument is still operational with no degradation in data quality.

Bourassa et al. (2014) merged SAGE II and OSIRIS data to examine trends in stratospheric ozone. Part of the apparent trends they showed in OSIRIS ozone has since been attributed to a time-dependent altitude registration offset and

has been corrected with a robust technique that uses stable features in the Rayleigh scattered limb radiance profile (e.g., Moy et al., 2017). The drift-corrected radiances have been used to process OSIRIS version 5.10 ozone. The 5.10 retrieval scheme is identical to that of version 5.07 and analysis of the improved OSIRIS data record results in reduced ozone recovery trends in the upper stratosphere (Bourassa et al., 2017). Comparisons of trends derived with OSIRIS Version 5.10 with OMPS USask 2D v1.0.2 (Zawada et al., 2017) ozone data records reveal the drift contained in OSIRIS version 5.07 has been

mitigated.

## 2.6   OMPS-LP

The Ozone Mapping and Profiler Suite Limb Profiler (OMPS-LP) on board the Suomi-NPP satellite has been taking measurements of limb-scattered sunlight from early 2012 to present (Flynn et al., 2006). OMPS-LP images the atmosphere using three vertical slits, one aligned with the orbital plane and the others separated by 250 km at the tangent point on either

side of the orbital track. Imaging allows OMPS-LP to obtain along track and vertical sampling of approximately 125 km and 1 km respectively. Spectral information in the range 270—1000 nm is obtained employing a prism spectrometer.

For HARMOZ, the OMPS-LP ozone data processed in the University of Saskatchewan are used (Zawada et al., 2017). Hereafter, we refer OMPS-LP to as OMPS for short, and the processor to as USask 2D v1.0.2. The USask 2D retrieval accounts for atmospheric variations along the orbital track by using the SASKTRAN-HR forward model (Zawada et

al., 2015) and simultaneously retrieving the ozone field for an entire orbit, rather than processing each vertical image separately. Only data from the center slit of OMPS is used as the other two slits are not aligned with the orbital track. Profiles are retrieved with a vertical resolution of $1-2$ km and an along track resolution of $300-400$ km. Individual profiles have a mean uncertainty of $4-6\%$ for most of the upper and middle stratosphere, with values increasing to approximately 30% just below the tropopause.

## 2.7   ACE-FTS

ACE-FTS is an instrument on-board the Canadian satellite SCISAT (Bernath et al., 2005; Bernath, 2017). It was launched in August 2003, and data are available from February 2004 to present. It provides latitudinal coverage from about 85°N to 85°S with complete coverage every 3 months. The ACE-FTS is a high-spectral-resolution ($0.02$ cm$^{-1}$) Fourier transform spectrometer measuring from 2.2 to 13 μm ($750 - 4400$ cm$^{-1}$) (Bernath et al., 2005). Operating in solar occultation

mode, ACE-FTS provides detailed profiles of the Earth's atmosphere for more than 30 chemical species.

The ACE-FTS processor employs a non-linear, least squares, global fit approach to retrieve volume mixing ratio profiles from spectra measured for each occultation using spectra simulated by a forward model. This processor is described





in Boone et al. (2005). The current version of the ACE-FTS data set used for HARMOZ is v3.5/3.6, as described in Boone et al. (2013). The only difference between v3.5 (typically 2004-Feb. 2013) and v3.6 (Mar. 2013-present and ~5% of 2004-Feb.2013 data) is the computer system used to perform the retrieval. A local computer was used for v3.5 and a shared supercomputing system is used for v3.6. For the ozone retrieval, 33 microwindows are used covering the range from 1027-

1169 cm$^{-1}$ with seven additional windows at 829, 923, 1105, 2149-2673 cm$^{-1}$ to better account for interfering species CFC-12, HCFC-22, CFC-11, $N_2O$, $CH_4$, HCOOH, $CO_2$, and their isotopologues. The altitude range of the retrieved ozone profiles is from cloud tops (~5 km) to 95 km and the vertical resolution is ~3-4 km (based on the field-of-view of the ACE-FTS instrument) (Boone et al., 2005).

Data quality flags are provided with the ACE-FTS dataset (Sheese et al., 2015). For HARMOZ, data at altitudes where
there was a flag value greater than 0 and data for all profiles with flag values of 4-6 were excluded (version 2.0 of the ACE-FTS v3.5/3.6 quality flags). In a recent validation study comparing ACE-FTS ozone to MLS and MIPAS profiles over the period 2004-2012, the average systematic bias was found to be +2% between 10 and 45 km and 0-+19% above 46 km (Sheese et al., 2017).

## 3    Preparation and selection of data for merging

For creating monthly zonal mean data from the individual instruments, 10° latitude bands from 90°S to 90°N are used. For all sensors, the monthly zonal average is computed as the mean of ozone profiles $x_k(z)$:

$$\rho(z) = \frac{1}{N} \sum x_k(z), \tag{1}$$

where $N$ is the number of measurements ($N>10$). The uncertainty of the monthly mean $\sigma_\rho^2$ can be estimated as the standard error of the mean:

$$\sigma_\rho^2 = \frac{s^2}{N}, \tag{2}$$

where $s^2 = \left\langle (x_k - \rho)^2 \right\rangle$ is the sample variance. Eq. (2) is valid for random samples of uncorrelated data. As shown by Toohey and von Clarmann (2013), some deviations of the real standard error of the mean from that calculated using Eq. (2) can be observed for satellite observations. In our study, Eq. (2) is used as an approximate estimate of the standard error of the mean, since no estimates considering the impact of the correlations caused by the orbital sampling are currently
available. In Eq.(2), we used a robust estimator for the sample variance: $s = 0.5 \cdot (P_{84} - P_{16})$, where $P_{84}$ and $P_{16}$ are the 84[th] and 16[th] percentiles of the distribution, respectively. Monthly zonal mean, sample variability $s$ and standard error of the mean from 15 to 50 km altitude in January 2008 are shown for the Ozone_cci instruments in Figure 2. The ozone distributions





shown in Figure 2 are very similar for all datasets. Due to large number of data available for averaging, the standard error of the mean is usually less than 1% in the stratosphere.

**Figure 2. Left: monthly zonal mean ozone profiles for January 2008 for Ozone_cci instruments, center: sample variability in %, right: standard error of the mean calculated using Eq. (2).**

Satellite measurements sample a continuous ozone field at some locations and times. To characterize the non-uniformity of sampling, we computed inhomogeneity measures in latitude, $H_{lat}$, and in time, $H_{time}$ (Sofieva et al., 2014a).



Each inhomogeneity measure $H$ is the linear combination of two classical inhomogeneity measures, asymmetry $A$ and entropy $E$ (for definition of these parameters, see Sofieva et al., 2014a):

$$H = \tfrac{1}{2}(A + (1 - E)).\qquad(3)$$

The inhomogeneity measure $H$ ranges from 0 to 1 (the more homogeneous, the smaller $H$). For dense samplers (MIPAS, SCIAMACHY, OMPS), the inhomogeneity is close to zero for nearly all latitude bins. For other instruments, inhomogeneity measure can be large for some latitude-time bins. The monthly zonal mean data with very high inhomogeneity are excluded from data merging (see below). The monthly zonal mean data from the individual instruments are available at the Ozone_cci web-page (www.esa-ozone-cci.org ).

For each instrument, latitude band and altitude level, the deseasonalized anomalies are computed as:

$$\Delta(t_i) = \frac{\rho(t_i) - \rho_m}{\rho_m},\qquad(4)$$

where $\rho(t_i)$ is the monthly mean value at a certain altitude and latitude band corresponding to time $t_i$ and $\rho_m$ is the mean value for the corresponding month $m$, i.e. $\rho_m = \dfrac{1}{N_m}\displaystyle\sum_{j=1}^{N_m}\rho_j$, $N_m$ being the number of monthly mean values $\rho_j$ in a given month $m$ available from all years.

For the Ozone_cci instruments, the seasonal cycle is evaluated using the overlapping period 2005-2011. The seasonal cycle for SAGE II is computed using years 1985-2004 and for OMPS using the years 2012-2016. In computation of deseasonalized anomalies, we ignored data from those latitude-time bins with the mean inhomogeneity $H_{\text{tot}} = 0.5(H_{\text{lat}} + H_{\text{time}}) > 0.8$ (for all considered satellite instruments, longitudinal inhomogeneity is negligible, Sofieva et al., 2014a).

Figure 3 shows examples of the seasonal cycles evaluated for the instruments considered. For instruments with coarse temporal and horizontal sampling, GOMOS and ACE-FTS, the seasonal cycle is evaluated less reliably. There are biases between instruments, which are in agreement with earlier validation and intercomparison results (Hubert et al., 2016; Rahpoe et al., 2015; Sofieva et al., 2017). Since the seasonal cycle for SAGE II and OMPS is evaluated using the periods different for that used for the Ozone_cci instruments, we added a corrected seasonal cycle (dashed lines in Figure 3):

$$\rho_{\text{SAGE,corr}}(m) = \rho_{\text{SAGE}}(m) \cdot \frac{\rho_{\text{SAGE},2002-2004}}{\rho_{\text{SAGE},1985-2004}} \cdot \frac{\rho_{\text{OSIRIS},2005-2011}}{\rho_{\text{OSIRIS},2002-2004}} \quad \text{and} \quad \rho_{\text{OMPS,corr}}(m) = \rho_{OMPS}(m) \cdot \frac{\rho_{\text{OSIRIS},2005-2011}}{\rho_{\text{OSIRIS},2012-2016}} \;,\; \text{where}$$

$\rho_{\text{instrument,time\_period}}$ is the average ozone for the indicated instrument and time period. (Note that this correction is not used in the merging algorithm, it is applied only for illustration in Figure 3.) The small overall biases with respect to SAGE II data are observed for OSIRIS and GOMOS (except in tropics at 20 km). Very good agreement between these instruments is also reported in several previous studies (Adams et al., 2013, 2014; Hubert et al., 2016; Kyrölä et al., 2013).





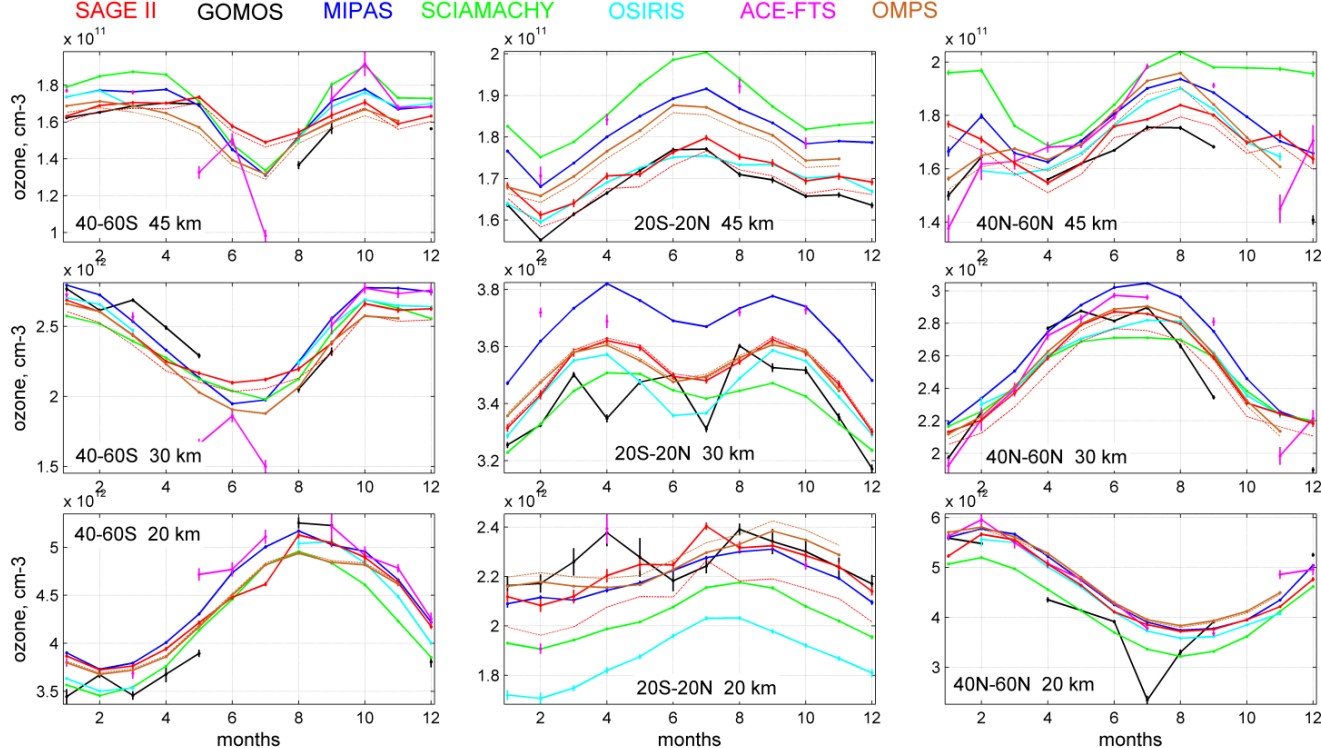

**Figure 3. Ozone seasonal cycle for latitudes 40°-60°S (left), 20°S-20°N (center) and 40°-60°N (right), for altitudes 45 km (top row), 30 km (center row) and 20 km (bottom row). Error bars are 2σ uncertainties according to Eq.(5). Dashed lines indicate corrected seasonal cycle for SAGE II and OMPS, see text for explanation. The seasonal cycles in the indicated zones are computed as the mean of seasonal cycles in 10° latitude bands.**

The uncertainty of the seasonal cycle value $\sigma_m$ for each month $m$ is evaluated from uncertainties of individual monthly mean values $\sigma_{\rho,j}$:

$$\sigma_m^2 = \frac{1}{N_m^2} \sum_{j=1}^{N_m} \sigma_{\rho,j}^2 \qquad (5)$$

It describes the propagation of the random uncertainties to the mean value. We would like to note that there is a significant interannual variability of ozone due to geophysical processes, (which can be characterized through a multiple linear regression with different proxies). These geophysical variations between different monthly values for a particular month are not characterized by Eq. (5).





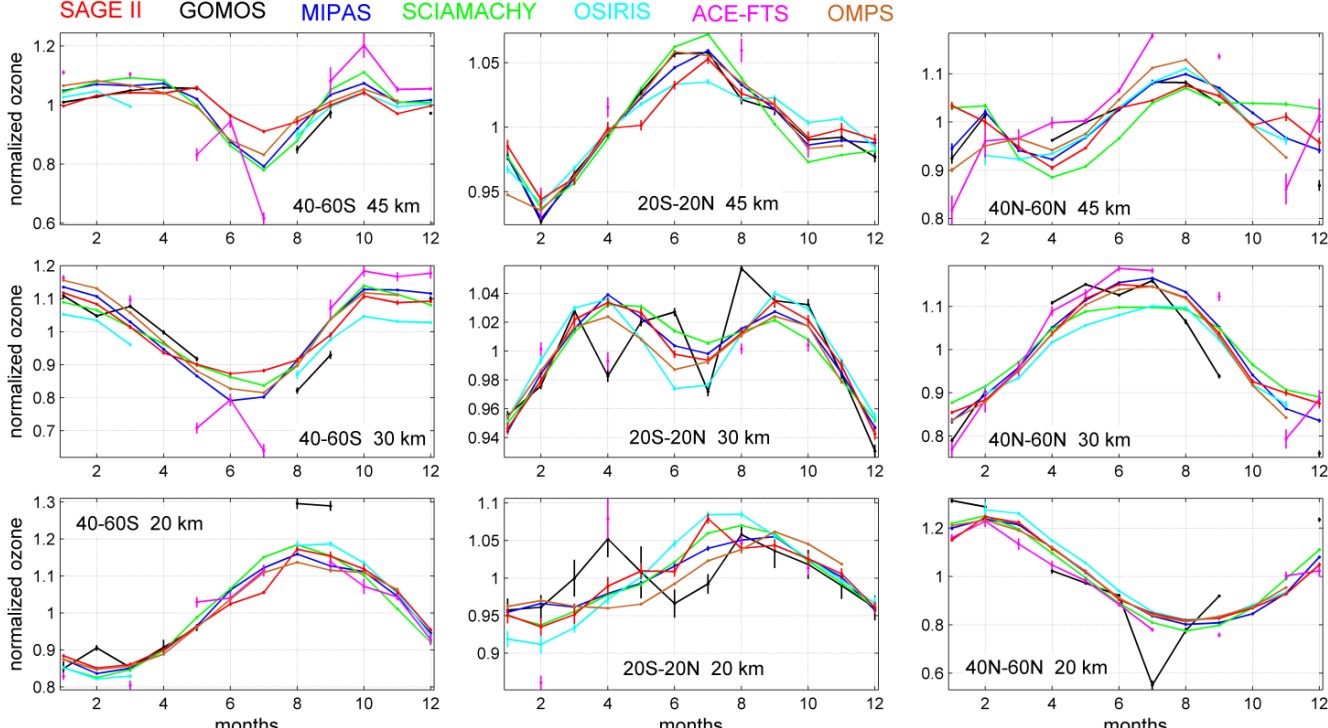

**Figure 4. Ozone seasonal cycle normalized on the mean ozone value for latitudes 40-60S (left), 20S-20N (center) and 40-60 N (right), for altitudes 45 km (top row), 30 km (center row) and 20 km (bottom row). Errorbars are 2σ uncertainties according to Eq.(5).**

5      The amplitude and phase of the seasonal cycle is very similar for SAGE II, MIPAS, SCIAMACHY, OSIRIS and OMPS, as illustrated in Figure 4, which shows the normalized seasonal cycle. The SCIAMACHY seasonal cycle is slightly different at 45 km at Northern mid-latitudes. This is caused by the altitude interpolation error, which is especially crucial at mid-latitudes in the upper stratosphere due to changing from semi-annual (above this altitude) to annual (below this altitude) variation regime. Due to a coarse altitude retrieval grid, SCIAMACHY data are more sensitive to the selected type of the

10     interpolation than other instruments. When using a log-linear interpolation, very similar results as for MIPAS are obtained in the upper stratosphere. SAGE II seasonal cycle in the middle and upper stratosphere in the latitude band 40°-60°S has significantly lower amplitude than those of other instruments, especially at 45 km, probably because of the sampling.

       After the removal of the seasonal cycle, the SAGE II deseasonalized anomalies are offset to the Ozone_cci mean anomalies in the years 2002-2005. The OMPS deseasonalized anomalies are offset to the mean Ozone_cci anomalies (which

15     are based on OSIRIS and ACE-FTS measurements in this period) in the years 2012-2016.

       Before merging, the deseasonalized anomalies of the individual instruments have been extensively inter-compared with each other by computing and visualizing the time series of difference of individual anomalies from the median anomaly. This method turns out to be a sensitive method for detecting an unusual timeseries behavior of the individual data



records. In particular, it was found that SCIAMACHY anomalies are larger in the beginning of the mission, for nearly all latitude bands and many altitude levels (see Figure 3 in the supplement, which shows the deviations of the SCIAMACHY deseasonalized anomalies from the median deseasonalized anomalies of SAGE II, GOMOS, MIPAS, SCIAMACHY, OSIRIS, ACE-FTS and OMPS). This might be attributed to possible pointing problems in the beginning of the mission;

5     therefore we decided not to use the SCIAMACHY data before August 2003 in the merged dataset. Similarly, OMPS anomalies are lower in the first three months of the mission (see Figure 6 in the supplement); this might be related to relatively coarse sampling of OMPS in the first three months of the mission and possible problems with pointing. Therefore, OMPS data were included in the merged dataset starting from April 2012, when the instrument operated in its full capacity.

10     **Figure 5. Deviations (in %, color) of deseasonalized anomalies for GOMOS, MIPAS, SCIAMACHY, OSIRIS, ACE-FTS, OMPS and SAGE II (indicated in the panels) from the median deseasonalized anomalies computed using all datasets, $\Delta_k - \Delta_{median}$. Latitude band is 30-40°S.**

After the data selection, the anomalies from individual instruments are found to be in good agreement with each other. This is illustrated in Figure 5, which shows the deviations of deseasonalized anomalies from each instrument relative to the





median anomaly for latitudes 30°-40°S. The deviations from the median anomalies are small, less than 5% for the majority of data and do not have statistically significant drifts with respect to the median anomaly (see also illustrations in the Supplement).

## 4    The merged SAGE II -Ozone_cci - OMPS dataset

5      We computed the merged anomaly as the median of the individual instruments anomalies, for each altitude level $z$ and for each latitude band $\theta$ and month $t$:

$$\Delta_{merged}(\theta, z, t) = \text{median}(\Delta_k(\theta, z, t)) , \tag{6}$$

where $\Delta_k$ indicates the individual instrument anomaly. Figure 6 illustrates the data merging: the upper panel shows the monthly zonal mean data, while the bottom panel shows individual anomalies and the merged (median) anomaly.

10    As observed in Figure 6, the biases between the individual data records are removed by computing the deseasonalized anomalies. In the merging, we filtered out individual anomaly values (locally for each latitude band and altitude level), which differ from the median anomaly more than 10% at latitudes 40°S-40°N and more than 20% in other latitude bands. This filtering is ineffective in the absolute majority of cases; it removes only a few exceptional anomalies from GOMOS and ACE-FTS, which are due to lower sampling.

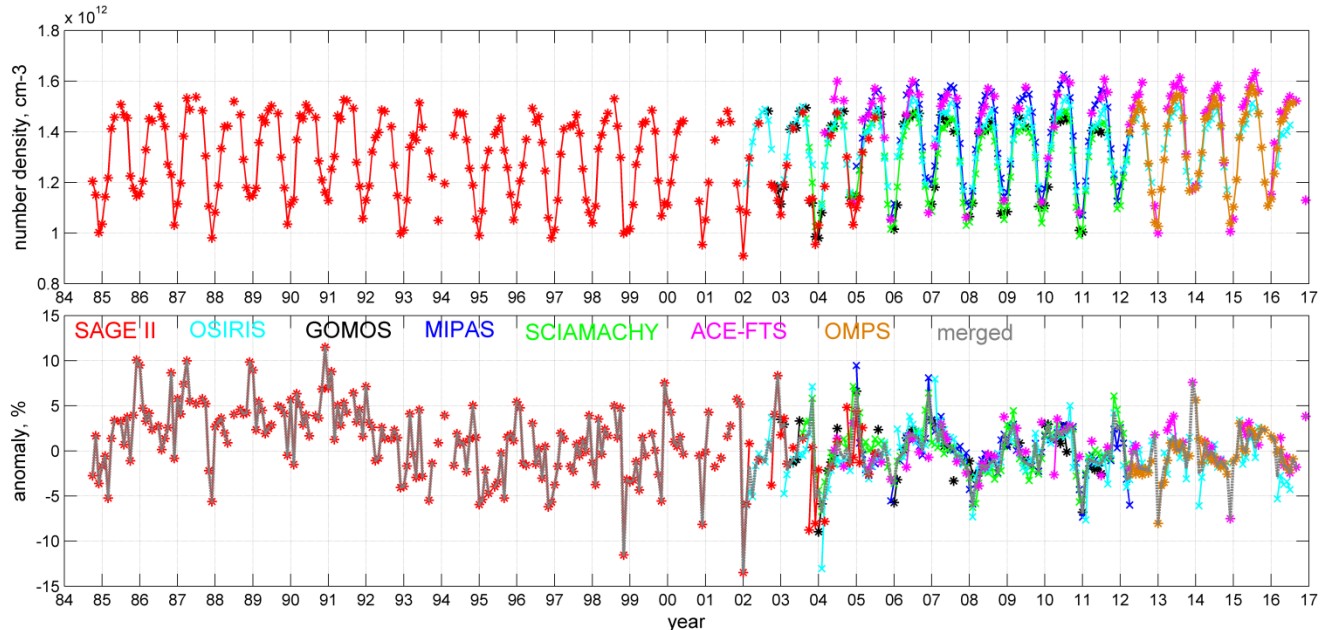

**Figure 6. Top: monthly zonal mean ozone at 35 km in the latitude band 40°-50°N. Bottom: individual deseasonalized anomalies and the merged anomaly (grey dashed line).**



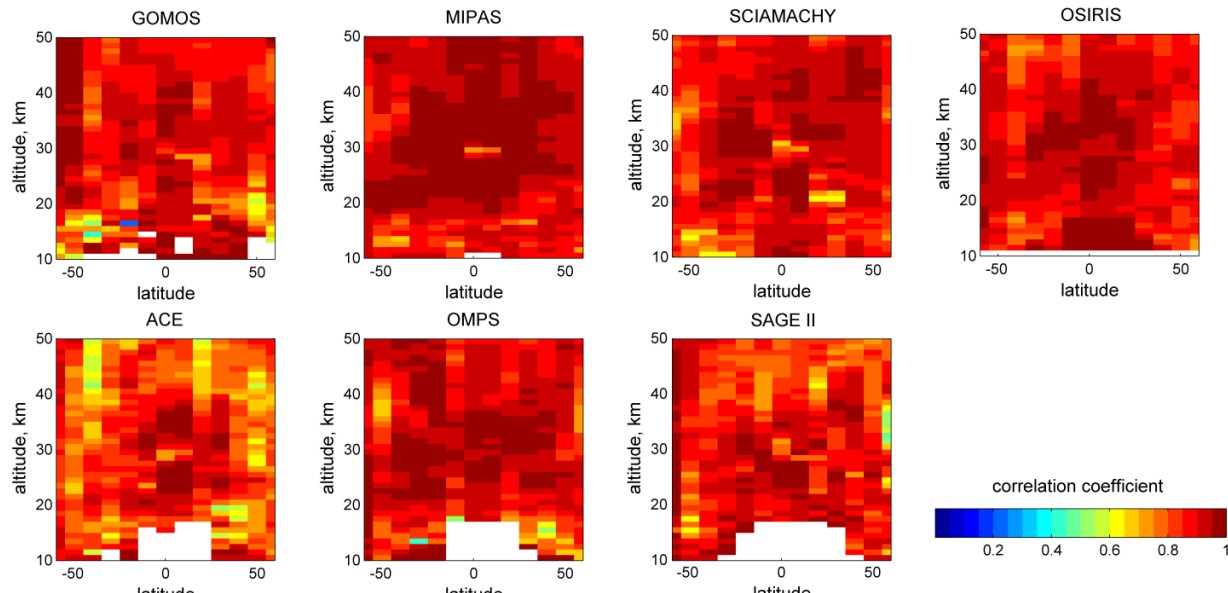

**Figure 7. Correlation coefficient between individual and merged deseasonalized anomalies in the period 2001-2016 at latitudes 60°S - 60°N.**

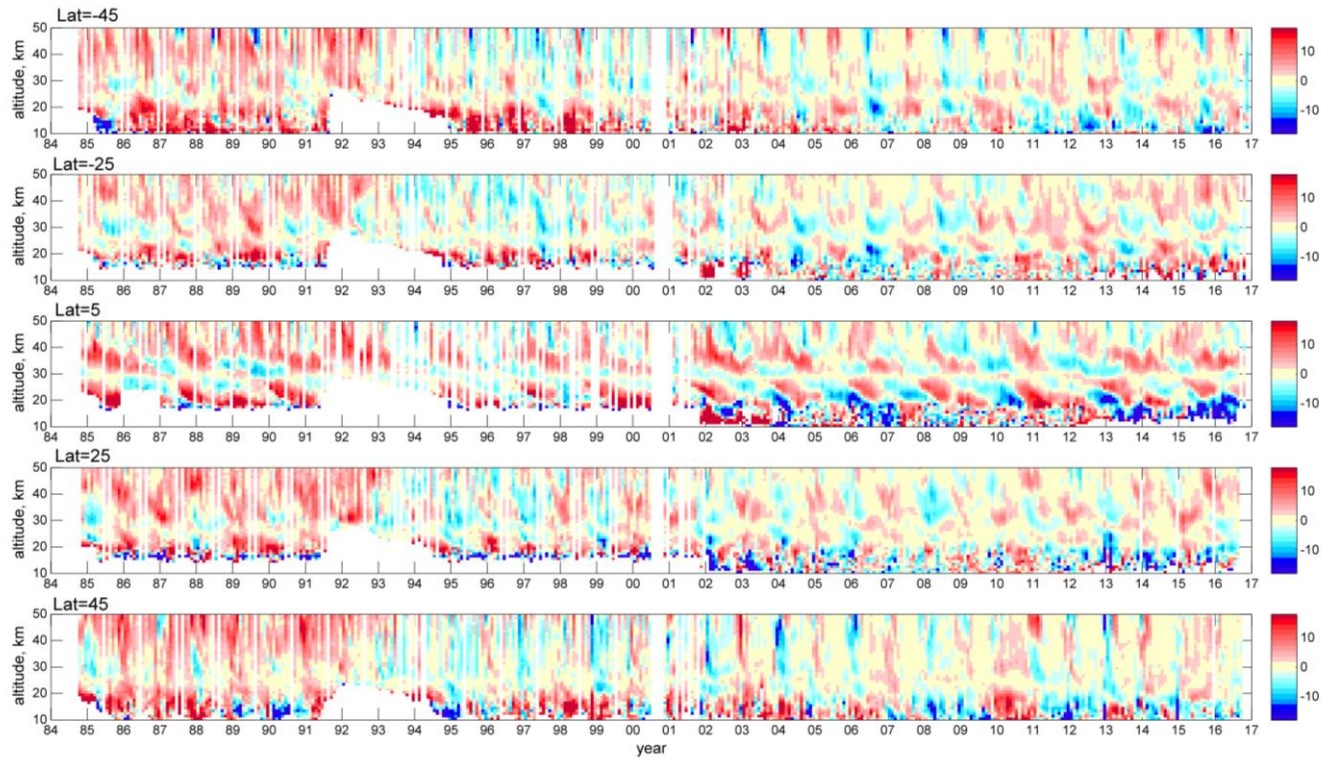

5    **Figure 8 Examples of merged deseasonalized anomalies (color: %), for several 10° latitude bands, centers of which are specified in the legend.**





An additional illustration of the very good agreement between the individual deseasonalized anomalies and the merged anomaly is presented in Figure 7, which shows the correlation coefficient between the merged anomalies and individual anomalies using the years 2001-2016. The correlation coefficient is above 0.9 for all instruments in most of the latitude bands and altitude levels.

Examples of merged deseasonalized anomalies for several latitude bands are shown in Figure 8. In the upper stratosphere at mid-latitudes a decrease from 1984 to 1995-1999 is observed, and then a gradual increase to the present. In the tropics, quasi-biennial oscillation (QBO) is observed.

The uncertainty of individual deseasonalized normalized anomalies (for each month) and each latitude-altitude bin $\sigma_{\Delta i}$ can be estimated as

$$\sigma_{\Delta i} = \frac{1}{\rho_m} \sqrt{\sigma_{\rho,i}^2 + \sigma_m^2} , \qquad (7)$$

where $\sigma_{\rho,i}$ is the uncertainty of the monthly zonal mean value Eq. (2), and $\sigma_m$ is uncertainty of the seasonal cycle from Eq.(5). We estimated the uncertainties of the merged deseasonalized anomalies (which correspond to median value) as

$$\sigma_{\Delta,merged} = \min\left( \sigma_{\Delta,j_{med}}, \sqrt{\frac{1}{N}\sum_{j=1}^{N}\sigma_{\Delta,j}^2 + \frac{1}{N^2}\sum_{j=1}^{N}\left(\Delta_j - \Delta_{merged}\right)^2} \right), \qquad (8)$$

where $\sigma_{\Delta,j_{med}}$ is the uncertainty of the anomaly of the instrument corresponding to the median value. Eq. (7) can be

interpreted as follows (see also Figure 12 in Appendix). If individual anomalies are significantly different, i.e., the corresponding error bars do not intersect (Figure 12, left), the uncertainty of the merged anomaly is the uncertainty corresponding to the median value. In case several instruments report a similar anomaly (intersecting error bars), this provides more confidence of this anomaly value, and the resulting uncertainty of the merged anomaly is approximated as

$\sqrt{\frac{1}{N}\sum_{j=1}^{N}\sigma_{\Delta,j}^2 + \frac{1}{N^2}\sum_{j=1}^{N}\left(\Delta_j - \Delta_{merged}\right)^2}$ . For example, in case of three coinciding anomalies with the same uncertainties

$\sigma$ , the uncertainty of the merged value will be $\sigma/\sqrt{3}$ . For the considered datasets, anomalies are usually very close to each other, so that several values are within the $\Delta_{merged} \pm \sigma_{\Delta,merged}$ interval, as illustrated by Fig.8 of the Supplement.

The uncertainty of the merged dataset is illustrated in Figure 9 for the same latitude bands as shown in Figure 8. As expected, the uncertainties in the time period when only SAGE II data were available are larger than uncertainties for time periods when several instruments have contributed. The average uncertainty is usually less than 4% before 2001 and below

1% for the years 2002-2017. In the UTLS, uncertainties are larger than in the stratosphere and are in the range of 3-9 %. At





mid-latitudes, uncertainties are larger in winter than in summer due to larger ozone variability during winter; this is observed clearly in the period before 2001.

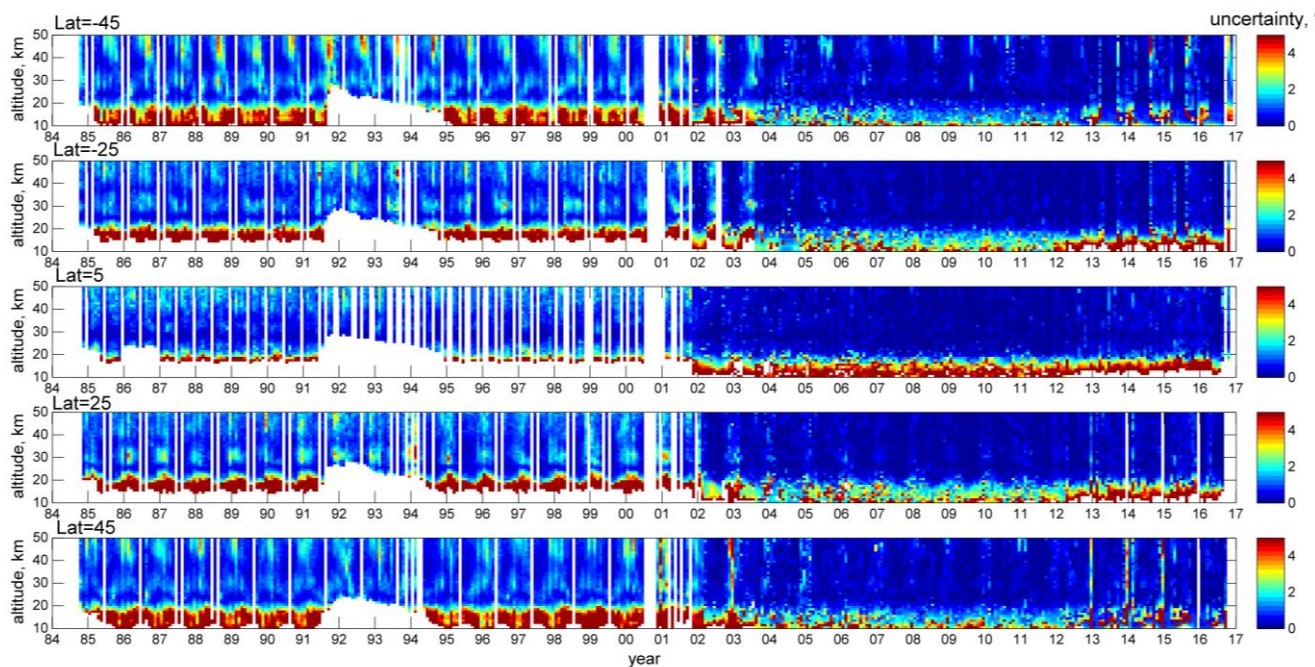

**Figure 9. Uncertainties of the merged deseasonalized anomalies in %, Eq.(8), for several 10° latitude bands, centers of which are specified in the legend.**

## 5   Ozone trends

The merged deseasonalized anomalies can be used directly for analyses of trends in the vertical distribution of ozone. For this purpose, a multivariate regression has been applied to the merged SAGE-CCI-OMPS data:

$$O_3(t) = PWLT(t,t_0) + q_1 QBO_{30}(t) + q_2 QBO_{50}(t) + s\,F_{10.7}(t) + d\,ENSO(t) ,\qquad (9),$$

where $PWLT(t, t_0)$ is a piece-wise linear term (constant and a hockey-stick trend with the turnaround point in 1997), $QBO_{30}(t)$ and $QBO_{50}(t)$ are the equatorial winds at 30 hPa and 50 hPa, respectively (http://www.cpc.ncep.noaa.gov/data/indices/), $F_{10.7}(t)$ is the monthly average solar 10.7 cm radio flux (ftp://ftp.geolab.nrcan.gc.ca/data/solar_flux/monthly_averages/), and $ENSO(t)$ is the 2 month lagged ENSO proxy (http://www.esrl.noaa.gov/psd/enso/mei/table.html). As shown by Kyrölä et al. (2013) and Laine et al. (2014), the best estimate of the turnaround point is in 1997 for majority of latitude bands and altitude levels. The sensitivity of regression results to the choice of turnaround point is discussed in Harris et al. (2015). Autocorrelations are removed using the Cochrane–Orcutt transformation (Cochrane and Orcutt, 1949).



Although uncertainties for the merged data are evaluated, they are not used in the regression analysis: different amounts of data available over time result in varying uncertainties over time (e.g., as shown in Figure 9), which might improperly weight the time series. In our regression, all data points are considered with equal weights, and the uncertainty of the fitted parameters is estimated from the regression residuals.

5      The regression is performed for each latitude band and for each altitude level independently. The regression is similar to that used in (Kyrölä et al., 2013). There are also other methods for evaluation of trends, e.g., using dynamical linear modelling (Laine et al., 2014). Since the main focus of our paper is generation of the merged SAGE-CCI-OMPS dataset, we selected a rather "standard" regression model (Eq. (9)). A short discussion on the sensitivity of regression results to the choice of the regression model is presented below.

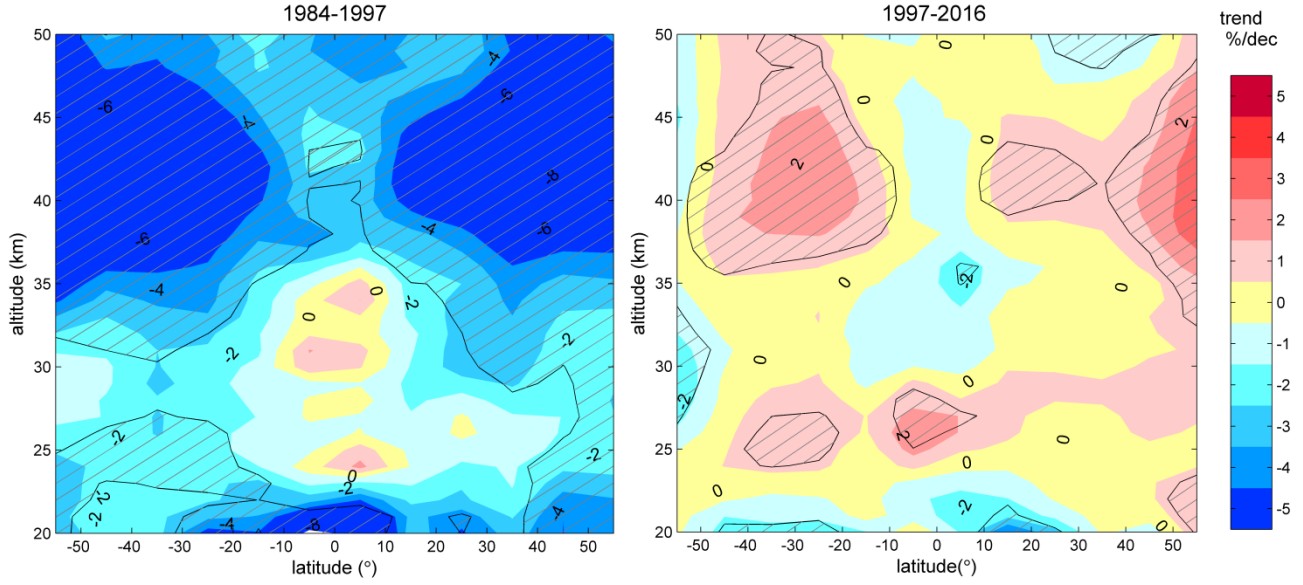

**Figure 10. The ozone trend in % per decade for different latitudes for 1984–1997 (left) and 1997-2016 (right). Shaded areas show regions where trends are statistically different from zero at the 95% level.**

The linear ozone trends before 1997 and after 1997 are shown in Figure 10. Shaded areas show regions where
15   trends are statistically different from zero at the 95% level. In the period 1984-1997, statistically significant negative trends from -4 %/dec to -8 %/dec are observed in the upper stratosphere. In the period 1997-2016, the ozone trends in the upper stratosphere are ~+2%/dec, and they are statistically different from zero at the 95% level in the extratropics. The ozone trends in large latitude bands, analogous to those studied in (WMO, 2014; Harris et al., 2015; Steinbrecht et al., 2017) are shown in Figure 11 (the anomalies in large latitude bands are created from 10°-latitude anomalies). The estimated ozone
20   trends are consistent with previous results (Bourassa et al., 2014; Harris et al., 2015; Kyrölä et al., 2013; Tummon et al., 2015; WMO, 2014) and indicate that the ozone recovery has started. A slightly different ozone trend analysis using the



SAGE-CCI-OMPS merged dataset by Steinbrecht et al. (2017) has resulted in nearly identical ozone trends (Figure 3 in Steinbrecht et al., 2017), which indicates only a weak dependence of ozone trend estimates on the regression method (and collection of proxies).

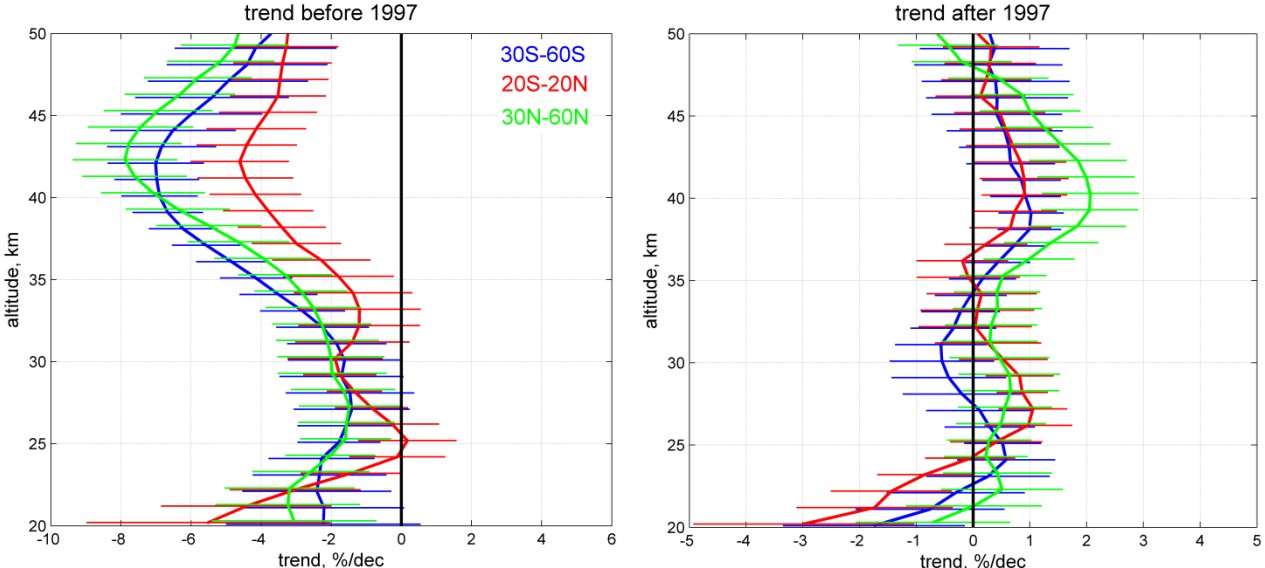

**Figure 11. Vertical profiles of ozone trends obtained by the multiple regression Eq.(9), in 1984-1997 (left) and in 1998-2016 (right), for broad latitude bands. Error bars are 2σ uncertainties.**

To study the sensitivity of ozone trend results to filtering of suspicious data, we created a version of the merged dataset in the same way as above but keeping the early periods of SCIAMACHY and OMPS operations and performed the same analysis. Keeping all data results in very minor changes in ozone trends after 1997, as illustrated by Fig.9 in the Supplement; the changes are mostly less than 0.3 %/dec. This is not unexpected, because the median estimate used in data merging is insensitive to outliers.

Although conversion of mixing ratio to number density for MIPAS and ACE-FTS is performed using the retrieved temperature profiles, this might introduce a minor inconsistency. To evaluate a potential effect on ozone trends, we created a version of the merged dataset, but without MIPAS and ACE-FTS data. This merged dataset contains only data that were retrieved in number density on a geometric altitude grid. Minor changes (<0.3 %/dec at latitudes 40°S-40°N, up to ~1%/dec at 40°-60°S and 40°-60°N) in ozone trends after 1997 are observed (Fig. 10 in the Supplement). The effect is rather small because MIPAS operated in 2005-2012, when other datasets are also available, and the coverage by ACE-FTS is limited. Furthermore, anomalies from individual datasets are in good agreement with each other, therefore significant changes in evaluated ozone trends are not expected.

The satellite data quality degrades in the UTLS. The merging principle seems to be optimal also for the UTLS, as it automatically removes biases, which can be significant in this region. The trends in the UTLS estimated using the merged



SAGE II – CCI – OMPS data follow the expected trend (declining ozone in the tropics just above the tropopause due to intensification of the Brewer-Dobson circulation). This is also in agreement with the dedicated studies on ozone trends in the tropical UTLS (e.g., Sioris et al., 2014).

## 6    Data availability and notes on data usage

The main dataset consists of the merged deseasonalized anomalies and their uncertainties described above. For the purpose of other applications (e.g., comparisons with models), we present also merged ozone concentration profiles. The details of computing merged number density profiles from the merged deseasonalized anomalies are presented in the Supplement. It is performed according to Eq. (4) by restoring the seasonal cycle. For trend analyses, it is recommended using the deseasonalized anomalies. According to the merging principle, the best quality of the merged dataset is in the stratosphere below 60° latitude.

The merged SAGE II – Ozone_cci – OMPS dataset is available from http://www.esa-ozone-cci.org.

## 7    Summary

We have presented the merged dataset of ozone profiles from several satellite instruments: SAGE II on ERBS, GOMOS, SCIAMACHY and MIPAS on Envisat, OSIRIS on Odin, ACE-FTS on SCISAT, and OMPS on Suomi-NPP. The merged dataset has been created with the aim of analyzing ozone trends in the stratosphere. For the merged dataset, we used the most recent retrieval versions of the satellite datasets. The datasets from the individual instruments have been extensively validated and inter-compared; only datasets, which are in good agreement and do not exhibit drifts with respect to collocated ground-based observations and with respect to each other, are used for merging.

The long-term SAGE-CCI-OMPS dataset is created by computation and merging of deseasonalized anomalies from individual instruments and associated uncertainties of the merged data are estimated. The merged SAGE-CCI-OMPS dataset consists of deseasonalized ozone anomalies in 10° latitude bands from 90° S to 90° N. The data are provided on altitude grid from 10 to 50 km covering the period from October 1984 to July 2016.

Ozone trends in the stratosphere are evaluated by applying a multiple linear regression to the merged SAGE-CCI-OMPS dataset. Negative ozone trends in the upper stratosphere before 1997 and positive trends after 1997 are observed. The upper stratospheric trends in the extratropics are statistically significant, and indicate onset of ozone recovery.

## 8    Appendix: Illustration of uncertainty of the merged deseasonalized anomalies

Here we present the illustration of Eq. (8). Two different cases are shown; left: individual anomalies (colored error bars) are significantly different and the corresponding error bars do not intersect; right: nearly coinciding individual anomalies. Since





the merged anomaly is simply the median value, in case of different anomalies (left), the uncertainty of the merged anomaly (black error bar) is the uncertainty corresponding to the median value (red in the considered example). If the number of instruments is even, the mean of the uncertainties corresponding to the nearest to the median values is taken. Several nearly coinciding anomalies from different independent instruments provide more confidence of this anomaly value, and the

resulting uncertainty of the merged anomaly is approximated as

$$\sigma_{merged} = \frac{1}{\sqrt{N}} \sqrt{\sum_{j=1}^{N} \sigma_{\Delta,j}^2 + \frac{1}{N} \sum_{j=1}^{N} \left( \Delta_j - \Delta_{merged} \right)^2} \qquad (10)$$

In the example shown in Figure 12 (right), the uncertainty of the merged value is close to $\sigma/\sqrt{3}$ ($\sigma$ is the uncertainty of each anomaly, the uncertainties are chosen to be the same for an illustration).

Eq.(10) can be also interpreted as follows. The expression under the square root represents the estimate of the standard

deviation for small samples (Ko et al., 2013), which, being divided by the square root of the number of measurements $\sqrt{N}$, gives the standard error of the mean.

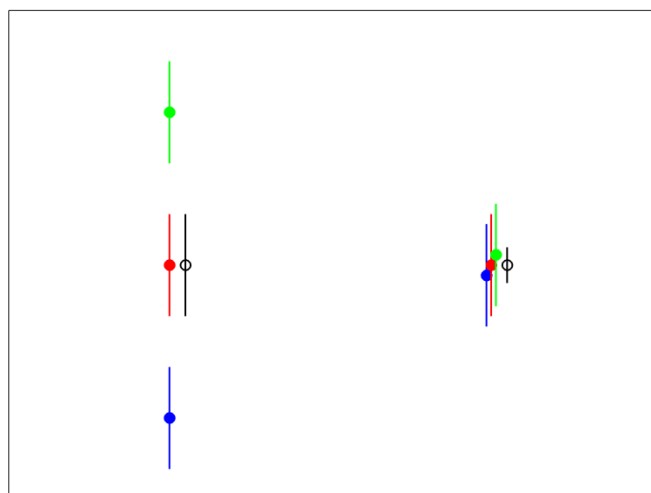

**Figure 12. Colored error bars: individual anomalies, black: merged (median) anomaly.**

**Acknowledgements**

The work is performed in the framework of ESA Ozone_cci project. The KIT team would like to thank the European Space Agency (ESA) for giving access to MIPAS level-1 data. The SCIAMACHY ozone retrieval was funded in parts by ESA, German Aerospace Agency (DLR), University and State of Bremen. The data set was calculated with resources provided by the North-German Supercomputing Alliance (HLRN). The GOMOS ALGOM2s dataset was created in the framework of

ESA ALGOM project. The FMI team thanks the Academy of Finland (INQUIRE project). The ACE mission is supported



primarily by the Canadian Space Agency. Odin is a Swedish-led satellite project funded jointly by Sweden (SNSB), Canada (CSA), France (CNES), and Finland (Tekes).

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
