# Peer review of "Merged SAGE II, Ozone\_cci and OMPS ozone profiles dataset and evaluation of ozone trends in the stratosphere"

_Atmospheric Chemistry and Physics, 2017_

## Referee Comment (RC1) · Anonymous Referee #1 · 3 Aug 2017

This paper presents a careful analysis of available long-term satellite ozone measurements to create a merged ozone data set covering the years 1984-2016. The work is carefully done and the authors demonstrate a strong understanding of the nuances of each of the separate data sets. The merging methodology and various statistical uncertainties are carefully explained in the manuscript, and the results seem reasonable to me. I agree with the choice of merging deseasonalized anomalies from the separate data. The merged data are used to estimate linear ozone trends before and after 1997, with the key result showing statistically significant ozone increases in the middle and upper stratosphere after 1997. I like the inclusion of various sensitivity tests in the text and the Supplement. Overall, this is a valuable project and I expect that these data

will be widely used in the research community for analyses and model comparisons. The paper is clearly written with appropriate references included, and I don't have any suggestions for changes. Well-done!

Very minor comments: p. 15, line 13: 'ineffective' doesn't sound right to me. How about 'The filtering does not affect the majority of cases, and only removes a few exceptional …...'. The caption to Fig. 12 could be improved to explain what is being shown, and what the y-axis represents.

---

## Referee Comment (RC2) · Anonymous Referee #2 · 8 Aug 2017

This manuscript introduces a new merged long-term data record of stratospheric ozone profiles from various satellite instruments based on their most recent retrieval versions. The dataset is used for ozone trend estimation from 1984-1997 and 1997-2016. Significant positive trends are found after 1997 in the extratropical upper stratosphere indicating the onset of ozone recovery. The authors present a very careful analysis of inter-sensor differences and uncertainties which are taken into account before combining the data. I appreciate that sampling inhomogeneities are also considered. The merging itself (as well as the trend estimation) are finally based on deseasonalized anomalies. Overall this study and the new data record provide a very valuable contribution to improve our understanding of long-term changes in ozone.

[Figure]

The manuscript is written in a clear and concise manner. I recommend publication and I have only a few very minor comments and technical corrections.

Minor comments:

p. 1, lines 30-31: "upper stratosphere" occurs twice in this sentence.

p. 1, lines 31-32: I would suggest to add that positive trends in the upper stratosphere are also expected due to decreasing temperatures in this altitude region.

p. 11, lines 14-26: The seasonal cycles for SAGE II and OMPS are computed using different reference time periods. For Figure 3 you calculate an adjusted seasonal cycle, but this is not used in the merging procedure. Might this have an impact on trend estimation ?

and p. 13, lines 13-15: I am not sure whether I understand this correctly. Are these offsets (SAGE II vs Ozone_cci from 2002-2005 and OMPS vs Ozone_cci from 2012-2016) due to the different reference time periods for the seasonal cycle ? And if so, do you correct for this ? Might these offsets have an impact on trend estimation ?

Technical corrections:

p. 24, line 5: missing journal

p. 24, line 35: remove curly brackets around "MIPAS"

p. 25, line 4: correct "H_{2}O" and "O_{3}"

p. 26, lines 13-15: refer to final revised paper

—————————————

---

## Author Response (AR1)

Dear Editor,

Thank you very much for your attention to our paper. We thank the reviewers for their very positive comments on our paper.  All their comments (a few minor corrections) are taken into account in the revised manuscript.

Please find below the response letters to the reviewers' comments (the same letters are posted in the interactive discussion). The revised manuscript with the modifications marked by "tracked changes" is also uploaded (the corrections in the references are not highlighted).

As a corresponding author, I confirm that all co-authors concur with the submission in its revised form.

Yours sincerely,

Viktoria Sofieva, Dr., Adj. Prof.
Finnish Meteorological Institute, Earth Observation
P.O. Box 503 (Erik Palmenin aukio, 1)
FIN-00101 Helsinki Finland
tel: +358 29 539 4698
fax: +358 29 539 3146
email: viktoria.sofieva@fmi.fi

**Review#1**

Dear Reviewer,

Thank you very much for your very positive comments on our paper acp-2017-598 "Merged SAGE II, Ozone_cci and OMPS ozone profiles dataset and evaluation of ozone trends in the stratosphere".

We rephrased the sentence p. 15, line 13 as you suggested: "This filtering does not affect the absolute majority of cases…"

We have added more explanation to the caption of Fig. 12. The new caption is: "Schematic illustration (arbitrary y-axis scaling) of two extreme cases of individual anomalies and their uncertainties. Colored error bars: individual anomalies, black: merged (median) anomaly. Left: significantly different individual anomalies with non-intersecting error bars, right: nearly coinciding individual anomalies."

**Review#2**

Dear Reviewer,

Thank you very much for your very positive comments on our paper acp-2017-598 "Merged SAGE II, Ozone_cci and OMPS ozone profiles dataset and evaluation of ozone trends in the stratosphere". All your comments are taken into account in the revised version.

Below we present the detailed replies to your comments. Your comments are in blue, replies are in black font.

Minor comments:

p. 1, lines 30-31: "upper stratosphere" occurs twice in this sentence.

Corrected

p. 1, lines 31-32: I would suggest to add that positive trends in the upper stratosphere are also expected due to decreasing temperatures in this altitude region.

Added.

p. 11, lines 14-26: The seasonal cycles for SAGE II and OMPS are computed using different reference time periods. For Figure 3 you calculate an adjusted seasonal cycle, but this is not used in the merging procedure. Might this have an impact on trend estimation ?

Yes, Figure 3 shows the adjusted seasonal cycle just for the illustration of seasonal cycles. The merging is performed on deseasonalized anomalies, in which instrument-specific seasonal cycle is removed (see also below).

and p. 13, lines 13-15: I am not sure whether I understand this correctly. Are these offsets (SAGE II vs Ozone_cci from 2002-2005 and OMPS vs Ozone_cci from 2012-2016) due to the different reference time periods for the seasonal cycle?

Yes, exactly.

And if so, do you correct for this? Might these offsets have an impact on trend estimation ?

Yes, these (small) offsets are corrected so that there are no jumps in the merged dataset. In the revised version, we added "As a result, small offsets in SAGE II and OMPS anomalies due to the different reference time periods for evaluation of the seasonal cycle are removed, and the anomalies from all instruments are aligned. "

Technical corrections:
p. 24, line 5: missing journal
p. 24, line 35: remove curly brackets around "MIPAS"
p. 25, line 4: correct "H_{2}O" and "O_{3}"
p. 26, lines 13-15: refer to final revised paper

Corrected (not highlighted with "tracked changes").